# Carbon Stock Prediction in Managed Forest Ecosystems Using Bayesian and Frequentist Geostatistical Techniques and New Generation Remote Sensing Metrics

Tsikai Solomon Chinembiri [1,*], Onisimo Mutanga [1] and Timothy Dube [2]

1. College of Agricultural, School of Agricultural Earth and Environmental Sciences, University of KwaZulu-Natal, Private Bag X01, Pietermaritzburg 3209, South Africa
2. Institute of Water Studies, Department of Earth Sciences, University of the Western Cape, Private Bag X17, Bellville 7535, South Africa
* Correspondence: chinembiri24500@alumni.itc.nl

**Abstract:** The study compares the performance of a hierarchical Bayesian geostatistical methodology with a frequentist geostatistical approach, specifically, Kriging with External Drift (KED), for predicting C stock using prediction aides from the Landsat-8 and Sentinel-2 multispectral remote sensing platforms. The frequentist geostatistical approach's reliance on the long-run frequency of repeated experiments for constructing confidence intervals is not always practical or feasible, as practitioners typically have access to a single dataset due to cost constraints on surveys and sampling. We evaluated two approaches for C stock prediction using two new generation multispectral remote sensing datasets because of the inherent uncertainty characterizing spatial prediction problems in the unsampled locations, as well as differences in how the Bayesian and frequentist geostatistical paradigms handle uncertainty. Information on C stock spectral prediction in the form of NDVI, SAVI, and EVI derived from multispectral remote sensing platforms, Landsat-8 and Sentinel-2, was used to build Bayesian and frequentist-based C stock predictive models in the sampled plantation forest ecosystem. Sentinel-2-based C stock predictive models outperform their Landsat-8 counterparts using both the Bayesian and frequentist inference approaches. However, the Bayesian-based Sentinel-2 C stock predictive model ($RMSE = 0.17$ MgCha$^{-1}$) is more accurate than its frequentist-based Sentinel-2 ($RMSE = 1.19$ MgCha$^{-1}$) C stock equivalent. The Sentinel-2 frequentist-based C stock predictive model gave the C stock prediction range of $1 \leq$ MgCha$^{-1} \leq 290$, whilst the Sentinel-2 Bayesian-based C stock predictive model resulted in the prediction range of $1 \leq$ MgCha$^{-1} \leq 285$. However, both the Bayesian and frequentist C stock predictive models built with the Landsat-8 sensor overpredicted the sampled C stock because the range of predicted values fell outside the range of the observed C stock values. As a result, we recommend and conclude that the Bayesian-based C stock prediction method, when it is combined with high-quality remote sensing data such as that of Sentinel-2, is an effective inferential statistical methodology for reporting C stock in managed plantation forest ecosystems.

**Keywords:** Bayesian methodology; classical geostatistics; multispectral remote sensing; carbon stock; plantation forest; managed ecosystem

## 1. Introduction

Plantation and natural forests serve as one of the main economic pillars in sub-Saharan Africa because they support both economic growth and human livelihoods. Due to the vulnerabilities caused by climate change, economies in Sub-Saharan Africa and other regions of the world must balance the need to protect the environment against the economic pressures brought on by population growth and poverty [1]. Nearly 50% of Africans live without access to electricity, and at least 60% of them still rely on wood for cooking and heating [2]. This disproportionate reliance on climate-vulnerable industries such as energy

and agriculture for economic survival and expansion leads to significant increases in the supply of primary energy and greenhouse gas emissions [3]. Forest biomass is regarded as a crucial part of monitoring forest resources according to the Food and Agriculture Organization's (FAO) 1994 International Forest Resources Monitoring Program [4]. Therefore, accurate monitoring and estimation of the forest's aboveground biomass (AGB) at the local and regional scales are essential for understanding how the forest's AGB contributes to the regional and global carbon cycles [5].

Zimbabwe is one of several African countries that, in their Nationally Determined Contributions (NDCs) to the Paris Agreement, have made bold proposals for establishing low-carbon and climate-resilient economies [3]. Yet Agriculture, Forestry and Other Land Use (AFOLU) still remain the biggest contributors of Greenhouse Gas Emissions (GHG), accounting for 54% of GHG in 2017 in Zimbabwe [6]. Deforestation resulting from agriculture expansion, increased stocking levels, the fetching of fuelwood, veld fires, the harvesting of timber for construction, mining, illegal settlements, tobacco curing, charcoal making, and commercial logging are some of the major drivers of GHG emissions in the AFOLU sector [7]. The Zimbabwean government also introduced the Greenhouse Gas Abatement Cost Model (GACMO) for establishing a GHG database [8]. The model can also be used as a tool for reporting, monitoring, and verifying transparency in mitigation actions for climate change.

The commonly used methods for estimating aboveground biomass (AGB) include mean biomass density, allometric equations, remote sensing, forest identity, geostatistics, and biomass expansion factors [9]. The method of statistical inquiry for any of these AGB estimation and prediction methods can be comprehensively categorised as either Bayesian or frequentist ones, depending on the circumstances of the investigation and assumptions underlying the inference. Notable differences between the Bayesian and the frequentist statistical methodologies regard the nature of the unknown parameters under investigation [10]. The frequentist paradigm treats parameters of interest as unknown and fixed ones, whilst the Bayesian framework regards all unknown parameters as uncertain ones, and therefore, should be characterised by a probability distribution [11].

The performance of Maximum Likelihood (ML), Least Squares and Bayesian approaches in Bogota, Columbia, was tested by Ghosh and Carriazo [12] in a hedonic estimation context and concluded that none of the aforementioned approaches are better than the other ones. However, because of the philosophical differences governing Bayesian and frequentist statistical techniques, the authors recommend the choice of the estimation technique to be grounded on the peculiarities of the policy problem at hand [12]. Some studies in the literature dwell at one of the two approaches as the principal methodology of inference. Notable work in the realm of biomass estimation using the Bayesian techniques include [13–16]. Recent studies assessing AGB distribution established accuracies of 17.52 Mg/ha [14] and 1.16 MgCha$^{-1}$ and 2.69 MgCha$^{-1}$ for Sentinel-2-based and Landsat-8-based Carbon (C) stock predictive models, respectively [16]. Other remote sensing- and machine learning-based efforts towards the estimation and prediction of AGB in recent times include those by Do et al. [17], who established mangrove AGB predictions in Vietnam at ranges from 6.51 to 368 Mgha$^{-1}$ and from 13.70 to 320.1 Mgha$^{-1}$ for remote sensing and Artificial Neural Networks, respectively.

Addressing a geostatistical research question from the Bayesian view point makes it possible to provide definitions of spatial predictors contributing to uncertainty in the unknown spatial covariance structure [18]. Kriging provided an optimal geostatistical technique under the frequentist paradigm that is employed in the description of spatial patterns and predicting values of a variable at unsampled locations and, consequently, evaluated the uncertainty associated with the predicted values [19].

Research making use of the frequentist geostatistical approach independent of the Bayesian technique for the estimation of C stock are also well documented in the literature. The authors of [20] mapped AGB in the Brazilian Amazon using Kriging with External Drift and established prediction accuracies for different sample sizes ranging from 0 to

110 for distances within 300 km radii from the prediction locations. The lowest RMSE for the estimated AGB for a sample size of 110 was 32.8 Mgha$^{-1}$, whilst the lowest accuracy for the lowest sample size of $n > 0$ was 48.06 Mgha$^{-1}$. To add to this, the authors of [21] predicted the AGB in a Wangyedia forest farm in China using Landsat-8 and the newly launched Landsat-9 datasets and arrived at RMSEs of 16.83 tha$^{-1}$ and 17.91 tha$^{-1}$, respectively. The authors of [22] coupled remote sensing-derived Sentinel-2 explanatory variables with geostatistics and machine learning algorithms in Mayanmar for predicting the aboveground biomass and established accuracies of 24.91 Mgha$^{-1}$ and 34.72 Mgha$^{-1}$ for Random Forest-based ordinary kriging and Random Forest-based co-kriging, respectively. As demonstrated by the results of Jiang et al. [21], Landsat-8 built AGB estimation models can still be superior to AGB models built from the successor Landsat-9 sensor, despite the relative spectral and radiometric improvements in the latest Landsat-9 sensor.

It is worthwhile to explore avenues for improving AGB prediction and estimation accuracy through the adoption of befitting statistical methods of inference for the production of high-quality reports applicable to climate change mitigation and climate change action. A significantly higher proportion of studies assessing the predictive performance of Landsat-8 and Sentinel-2 in aboveground biomass estimation favour Sentinel-2 over Landsat-8, though the differences in prediction performances are insignificant [21–23]. On the other hand, studies that have employed the Bayesian spatial hierarchical technique coupled with remote sensing-derived ancillary data have always outperformed similar studies conducted using the frequentist approach for predicting the AGB [14,24–27].

From the frequentist viewpoint, the interpretation of a confidence interval (CI) is hypothesised based on Neyman understanding, where the CI gives a measure of uncertainty by taking into account the long-run frequency of replicated experiments [28,29]. This suggests that if a practitioner/forester gathers 500 datasets on tree *dbh* from independent trials for estimating the parameter of a C stock prediction model and constructs a, with a 97% confidence interval for the parameter estimate for each dataset, at least 97 of the CIs would be expected to contain the true (but fixed) unknown model parameter [28,29]. However, in most practical settings, foresters or practitioners would not have access to multiple datasets and usually have a single dataset, as it is rather costly and unfeasible to undertake multiple experiments. The constructed CI may or may not contain the true (but fixed) unknown model parameter. On the other hand, the Bayesian credibility interval is a more pragmatic proposition as the credible interval is built in a manner that guarantees that there is a certain probability associated with getting the true (but random) unknown model parameter [30,31]. By the same reasoning, if a practitioner estimates the C stock predictive model using a single tree *dbh* dataset and constructs a, with a 97% credible interval, there would be a 97% probability that the true (but random) unknown model parameter is contained within that credible interval [32].

Since uncertainty is inherent in spatial prediction problems at unsampled locations, the Bayesian approach handles it better than the frequentist approach does because it benefits from having access to the full posterior predictive distribution of the modelled variable [31,33,34]. The principled way in which Bayesian inference incorporates pre-experimental information in the form of priors and experimental data, combined with the real-world benefits of parameters of carbon accounting and climate change action, makes it worthwhile to compare the two inferential paradigms when one is using freely available and new generation remote sensing data. As a result, the current study is an extension of the current earth observation-based inferential techniques for C stock accounting in climate change adaptation and mitigation within the United Nations Framework Convention on Climate Change (UNFCCC). As previously hypothesised, we set out to determine whether the Bayesian inferential approach can handle the uncertainty inherent in spatial prediction phenomena better than its frequentist counterpart can.

## 2. Methods

### 2.1. Study Area

We undertook the study at Lot 75A of Nyanga Downs in the Manicaland province of the eastern highlands of Zimbabwe. The area of interest is in the Nyanga district of the aforementioned province and has dominant tree species comprised of *Pinus patula*, *Eucalyptus grandis*, and *Eucalyptus camaldulensis*. Dotted patches of the area lying between latitude 32°40′ E and 32°54′ E and longitude 18°10′12″ S and 18°25′4″ S as illustrated in Figure 1 have undergone changes in land use and are currently used for gold panning, agriculture, and grazing [35].

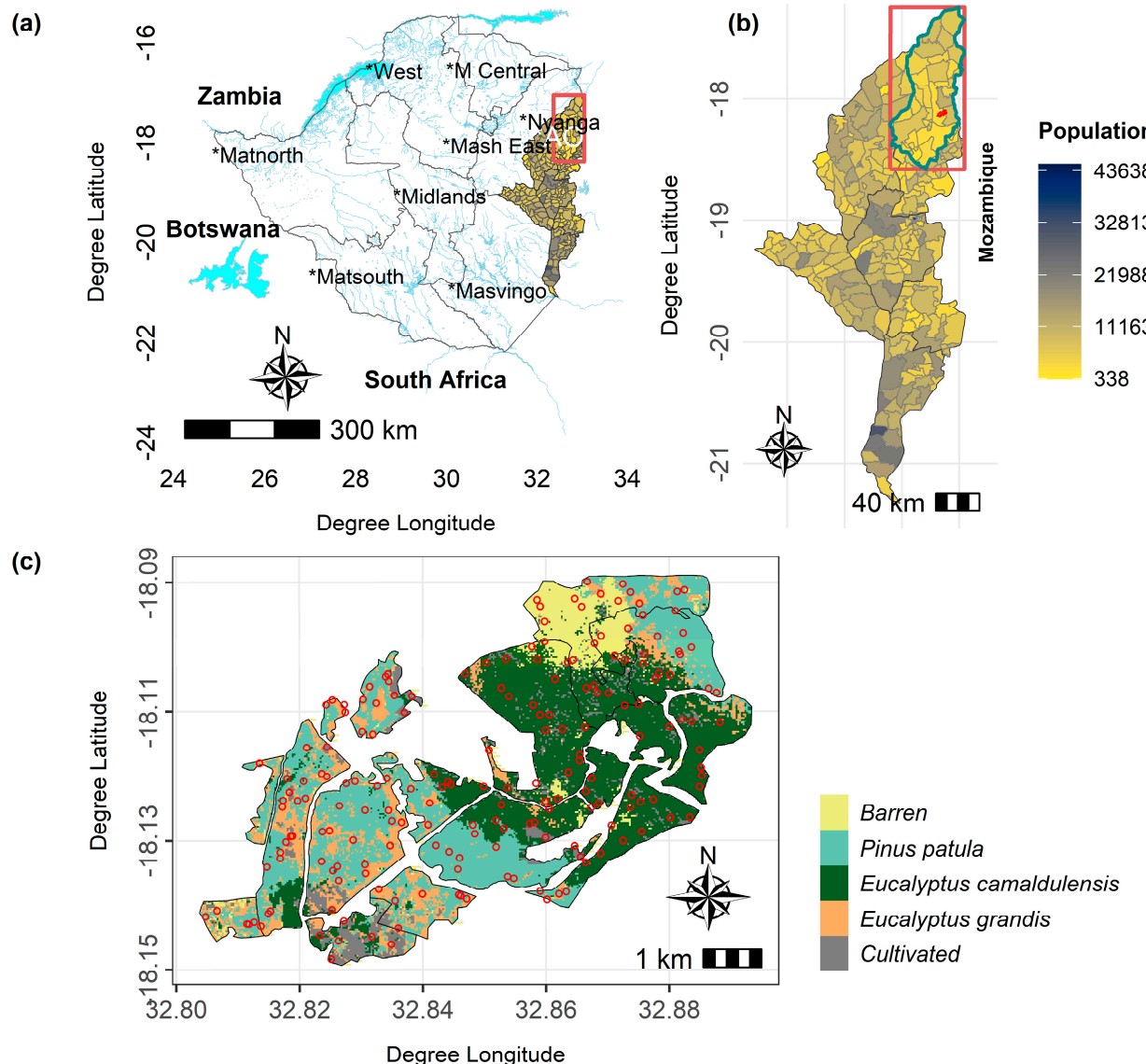

**Figure 1.** Study map showing (**a**) location of the province where samples for the study were collected, (**b**) study area location within the sampled province, and (**c**) plantation forest species distribution in the area of interest (AOI). * Shows provincial names.

Changes in land use to grazing and agriculture came after some parts of the commercially owned plantation forests were redistributed by the government of Zimbabwe to communal farmers in 2000. This development removed access barriers that were in place within the commercial plantation forests before the land redistribution, giving settlers more access to forest resources [36]. The area of study which covers an approximate area of

2767 ha, receives variable amounts of rainfall ranging from 741 mm to 2997 mm, and a mean annual precipitation of 1200 mm [37]. Annual mean temperatures are also varied, with a minimum annual range from 9 °C to 12 °C and a maximum range from 25 °C to 28 °C. Extensive wild fires occur at high altitudes due to the hot weather experienced during the summer season of the year between the months of August and November [38]. Rapid population growth (Figure 1b) within the jurisdiction of the studied region is also a possible factor driving rapid land use change, leading to veld fires and logging for opening up more land for agriculture.

### 2.2. Remote Sensing Covariates

The relatively low cost of acquiring Landsat and other freely available earth observation sensors, in addition to their spatial coverage, makes them indispensable for natural resource modelling. The recently launched Sentinel-2 satellite sensor is made up of six bands, which can be compared to the Landsat-8 bands, and also carries three additional bands comprising the red edge (RE) spectrum [39]. The red edge bands are positioned at 704, 740, and 782 nm, whose band widths are 15, 15, and 20 nm, respectively. The red edge forms the major spectral feature of vegetation located between the high reflectance in the NIR (750 nm) and the red absorption maximum (680 nm) [40]. Sentinel-2′s surface bands have 10 m and 20 m spatial resolutions as compared to Landsat-8′s 30 m bands. These differences in sensor configuration and properties form the basis for assessing their mapping accuracies.

### Landsat OLI and Sentinel-2 MSI Imagery

We obtained Landsat-8 images from the United States Geological Survey Earth Explorer site as georeferenced and analysis ready data (ARD) (http://earthexplorer.usgs.gov, accessed on 13 February 2023). We filtered the datasets for cloud cover and set cloud shadow thresholds to below 10%. We downloaded Sentinel-2 cloud free images on the 20 September 2020, coinciding with the time that we acquired the Landsat-8 data, which covered the whole area of interest, including Lot 75A in the Nyanga Downs in the eastern highlands of Zimbabwe. Sentinel-2 imagery with 13 spectral bands was acquired as level-1C 12-bit fixed Top of the Atmosphere (TOA) reflectance values. We carried out the orthorectification and pre-processing of the Sentinel-2 level 1-C data using the *sen2r* package in the R Statistical and Computing Environment [41].

We derived the Enhanced Vegetation Index (*EVI*), Soil-Adjusted Vegetation Index (*SAVI*), and Normalised Difference Vegetation Index (*NDVI*) from each of the two sensors as independent variables for use in C stock prediction in a managed plantation forest in Zimbabwe. The authors of [40,41] employed the aforementioned vegetation indices as independent variables in AGB estimation and assessment. We therefore employ these variables using philosophically different statistical research methodologies in order to assess the best framework that can be applied in C accounting and reporting for climate change studies.

### 2.3. Sampling Design
Spatial Coverage Sampling and Mapping of Regionalised Variables

We carried out sampling in an area that had not been sampled before, and hence, the scales of spatial variability were not known beforehand. Under such circumstances, the Mean Squared Shortest Distance (MSSD) is a befitting objective function, which we utilised in order to optimize the sample locations. We utilised the *k-means* clustering algorithm for uniform area coverage sampling. According to Walvoort, Brus, and de Gruijter [42], the even distribution of sampling locations within a study domain can enhance the mapping and estimation of regionalised variables. This assertion is further confirmed by Brus, de Gruijter, and van Groenigen [43], who demonstrated how even coverage of the study area with sampling observations can be utilised for the dual role of estimating the spatial means of regionalised variables and resolving mapping in forestry, soil, and environmental

research. The MSSD remains a dominant methodology for optimizing the sampling pattern over other methods such as Spatial Simulated Annealing (SSA) for both prediction and estimation designs for regionalised variables. The suitability of this design for locations where sampling schemes cannot be extended beyond a single phase is well documented.

We subdivided the study domain (*D*) into compact subunits through the clustering of building blocks making up the sampling domain using the *k-means* optimization procedure [44,45]. *x* and *y* coordinates of the central points of the building blocks are the classification variables for the *k-means* optimization function. We utilised the centroids of clusters as sample locations, where the sampling plots for C stock were set up.

### 2.4. Carbon Stock Data

### 2.4.1. Aboveground Tree Biomass (AGTB) Field Measurement

Measurements of all trees with at least 10 cm diameter at breast height (*DBH*) (at 1.3 m) were taken using 500 m$^2$ circular plots from the 19 September to the 24 October 2021. Diameter and linear tapes were used for the tree measurements, and trees with dbh less than 10 cm were excluded as they are generally regarded to have an insignificant C stock [3]. As the average slope within the study area was generally less than 30%, we did not consider slope correction for the measured outcome variable [46]. We carried out optimization of the resulting 200 sampling points using the ***spcosa*** package implemented using the R Statistical and Computing Environment [44,45]. We pre-uploaded the 200 probable sampling observations into a 72 H handheld Garmin GPS before setting out for the field work programme. The number of actual sampling points of forest biomass obtained during the field exercise was one hundred and nine, as nine of the pre-loaded sampling locations fell outside the boundaries of the defined study domain (Figure 1).

### 2.4.2. Biomass Calculation and Derivation of C Stock

Allometric equations used by Brown [7] were applied in the calculation of AGB *Pinus* species, whilst the AGB of *Eucalyptus* species was calculated using allometric equations developed by the authors of [47]. The same allometric equations used for *Pinus* and *Eucalyptus* species were also applied to *Eucalyptus* and *Pinus* species of the Manica province in Mozambique, whose climatic and weather conditions largely resemble those of the studied region in the eastern highlands of Zimbabwe. We then converted the AGB of every individual tree to C stocks per species through the conversion factor in [4]. Estimated per plot AGB values were then expanded to a standardised unit area of a hectare measured in MgCha$^{-1}$.

### 2.5. The Bayesian Geostatistical Modelling Framework

The modelling framework for the Bayesian and frequentist geostatistical approaches using new generation remote sensing Landsat-8 and Sentinel-2 as data sources is illustrated in Figure 2. Both approaches culminate in C stock predictions whose qualities were evaluated using cross-validation statistics (Figure 2). We assumed the Bayesian hierarchical methodology in order to have a full account of the parameter uncertainty for the measured C stock as given in Equation (1) [48].

$$Y(s) = X^T(s)\beta + w(s) + \varepsilon(s) \tag{1}$$

where:

$w(s)$ represents the spatial random effects term;

$\beta$ denotes a vector of covariate coefficients;

$X^T(s)$ denotes a vector of predictors measured at the same location as $Y(s)$;

$Y(s)$ denotes the sampled C stock variable;

$\varepsilon(s)$ denotes the white-noise-assumed independent and identically distributed (*i.i.d.* $N(0, \sigma_\varepsilon^2)$).

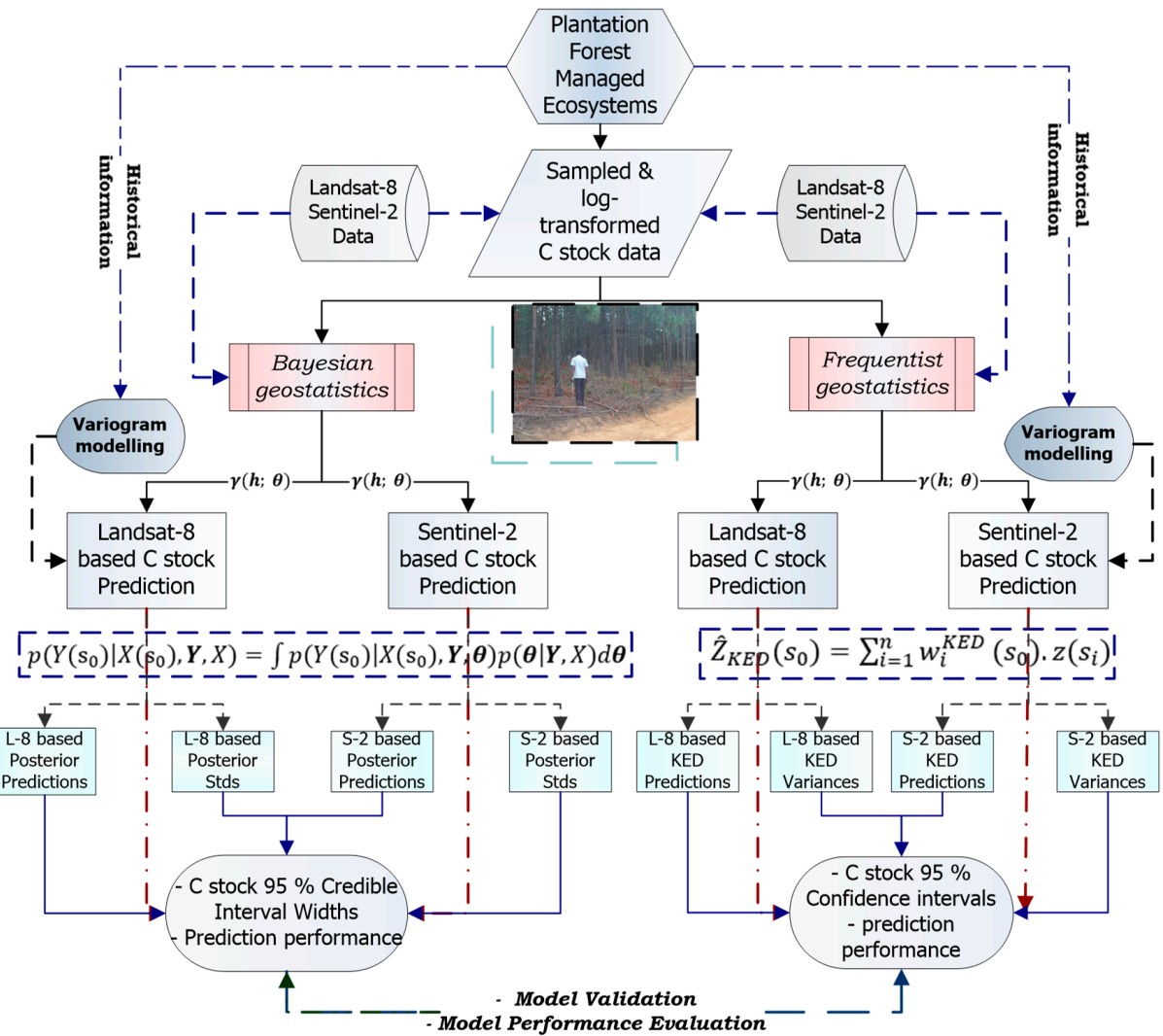

**Figure 2.** Bayesian and frequentist geostatistical modelling framework.

We performed the simultaneous estimation and prediction of C stock parameters by making use of the Markov Chain Monte Carlo (MCMC) technique to derive and calculate C stock predictions at unvisited locations, as in Equation (2).

$$\hat{Y}(s)' = X^T(s)'\hat{\beta} + \hat{w}(s)' \tag{2}$$

All hierarchical models were made using the *spBayes* package (Finley, Sudipto, and Carlin [49]) in the R Statistical and Computing Environment [50]. According to Gelfand [51], the vector of model parameters, $\boldsymbol{\theta} = \beta, \sigma^2, \phi, \tau^2$, is treated as random and mutually independent variables and is assigned prior distributions. We therefore sampled the posterior distribution of the parameters of interest, $\boldsymbol{\theta}$, as in Equation (3).

$$p(\boldsymbol{\theta}|\boldsymbol{y}, X) \propto p(\boldsymbol{\theta}) \times N(\boldsymbol{w}|0, \Sigma_w) \times N(\boldsymbol{y}|X^t\boldsymbol{\beta} + \boldsymbol{w}, \Sigma_\varepsilon) \tag{3}$$

Equation (3) was employed in the quantification of uncertainties in model parameters and C stock predictions at unsampled locations derived using Equation (4).

$$p(y_0|\boldsymbol{y}, \boldsymbol{X}, \boldsymbol{x}_0) \propto \int p(y_0|\boldsymbol{y}, \boldsymbol{\theta}, \boldsymbol{x}_0) p(\boldsymbol{\theta}|\boldsymbol{y}, \boldsymbol{X}) d\boldsymbol{\theta} \tag{4}$$

where:

$y_0$ represents the predicted C stock at a site $s_0$ and $x_0$ are the predictor values at site $s_0$.

The overall mean of the sampled C stock was assigned a normal prior, whilst the regression coefficients were assigned a multivariate normal prior. Since the study made use of ancillary data from Landsat-8 and Sentinel-2 sensors, we specified two classes of priors for hierarchical modelling of the outcome variable. We assigned an inverse gamma distribution for the C stock data and measurement error variance, whilst the spatial decay parameter, $\phi$, was assigned a uniform prior as indicated by $p(\boldsymbol{\theta}_1)$ and $p(\boldsymbol{\theta}_2)$ for the Landsat-8 and Sentinel-2 sets of priors, respectively. The assignment of the prior distribution on the spatial decay parameter was guided by the maximum distance between the sampling locations (2413 m) within the geographic domain ($\boldsymbol{\theta}_1, \boldsymbol{\theta}_2 \in D$) of the studied region.

As we expected the white noise error variance (nugget, $\sigma_\varepsilon^2$) to be smaller than the structured variance was, $\sigma_w^2$, scale parameter values were adopted in order to express the preference that $\sigma_\varepsilon^2 < \sigma_w^2$ [52]. A uniform prior with support covering the geographic domain of the study area was assigned to the spatial decay parameter, $\phi$. Prior distributions on the modelled parameters were derived from covariance parameters from the exploratory variograms of the two multispectral remote sensing sensors. A Metropolis–Hastings algorithm for MCMC was utilised [48]. We then specified an algorithm of one chain comprised of 20,000 MCMC iterations for the posterior densities of the model parameters. Fifteen thousand chains were discarded as burn-in.

Bayesian Model Validation and Diagnostic Evaluation

We compared the spatial model, the spatial-intercept-only model, and the independent error model (simple multiple linear regression) for assessing the performance of predictions from the Bayesian hierarchical modelling approach with the Uniform (***Unif***) and Inverse Gamma (***IG***) priors on the spatial decay and the spatial random effects as illustrated for $\theta_1$ and $\theta_2$, respectively.

$$p(\boldsymbol{\theta}_1) = \boldsymbol{Unif}(\phi | 0.38, 0.0012) \ \times \ \boldsymbol{IG}\left(\sigma^2 \middle| 0.52, 1.58\right) \ \times \ \boldsymbol{IG}\left(\tau^2 \middle| 0.1, 1.58\right) \ \times \ \boldsymbol{MVN}\left(\beta \middle| 0, \Sigma_{\boldsymbol{\beta}}\right)$$

$$p(\boldsymbol{\theta}_2) = \boldsymbol{Unif}(\phi | 0.38, 0.0012) \ \times \ \boldsymbol{IG}\left(\sigma^2 \middle| 0.052, 0.0028\right) \ \times \ \boldsymbol{IG}\left(\tau^2 \middle| 0.1, 1.52\right) \ \times \ \boldsymbol{MVN}\left(\beta \middle| 0, \Sigma_{\boldsymbol{\beta}}\right)$$

The predictive performance of each of the three models was tested using a *k-fold* cross-validation algorithm, which performed the cross-validation through random splitting of the measured 191 C stock observations into approximately ten equally seized segments [53]. We calculated validation metrics of the Mean Absolute Error (MAE), Root Mean Square Error (RMSE) and other goodness-of-fit statistics such as the Deviance Information Criterion (DIC) for ranking candidate models on their ability to fit data [54]. Desirable and better fitting models display lower values of the DIC, whilst better performing models would have the lowest $k(k = 10)$-fold MAE and RMSE. We also assessed the conformance of the models to the modelling assumptions using graphical diagnostic plots of model residuals [15,53].

### 2.6. The Frequentist Geostatistical Modelling Framework

### 2.6.1. Carbon Stock Spatial Interpolation

We utilised ordinary kriging (OK) as a baseline for assessing how covariates derived from Landsat-8 and Sentinel-2 impact the C stock model using a frequentist geostatistical methodology. Covariates are a means of enhancing the predictive properties of spatial models [55]. The manner in which auxiliary variables lead to better kriging estimates than the ordinary kriging algorithm can produce is shown by the work in [54,56]. We therefore employed kriging with External Drift (KED) as a geostatistical methodology for modelling the spatial distribution of C stock in managed plantation forest ecosystems using vegetation indices derived from Landsat-8 and Sentinel-2. As Equation (5) shows, the KED algorithm restricts stationarity within a search neighbourhood and provides more detailed information compared to that achieved by ordinary kriging [57]. The standard measurement unit for forest C stock accounting is a hectare [3], and hence, we utilised a 100 m $\times$ 100 m resolution grid for kriging, with the normality assumptions tested with residuals of the selected liner model [58].

$$Z_{KED}^*(\mu) = \sum\nolimits_{\alpha=1}^{n(\mu)} \lambda^{KED}(\mu) Z(\mu_\alpha) \tag{5}$$

where:

$Z^*_{KED}(\mu)$ denotes the KED estimated value at site $\mu$;

$\lambda^{KED}(\mu)$ denotes the KED weights pertaining to $n$ samples at site $\mu$;

$Z(\mu_\alpha)$ denotes the sample values inside the search neighbourhood at alpha.

### 2.6.2. Frequentist Model Validation and Diagnostics

We followed the method of cross-validation as outlined in [59] for assessing the quality of predictions of each of the C stock predictive models constructed from Landsat-8 and Sentinel-2 satellite sensors. We therefore presented the validation statistical metrics in the form of RMSE, MSE, ME, and the Predicted Residual Sum of Squares (PRESS).

### 2.7. Variogram Modelling of the Regionalised Variable

The Bayesian and frequentist geostatistical modelling philosophies both make use of the variogram as the basis for establishing spatial covariance parameters. We presented the variogram of residuals from the linear modelling of C stock with predictors from both Landsat-8- and Sentinel-2-derived vegetation indices. Variogram modelling was, therefore, utilised as a basis for assessing the robustness of the spatial correlation structure of the modelled C stock response variable [60–62]. The regionalised variable was also transformed into a logarithmic scale to ensure conformance to the normality of residuals modelling assumptions using the Box–Cox transformation technique [63].

## 3. Results

### 3.1. C Stock Descriptive Statistics

Descriptive statistics of the sampled C stock data relating to the measured forest parameters are illustrated in Table 1. The mean C stock for *Eucalyptus camaldulensis*, *Eucalyptus grandis*, and *Pinus patula* species were 2485.3 MgCha$^{-1}$, 405.7 MgCha$^{-1}$, and 377.9 MgCha$^{-1}$, respectively. *Eucalyptus camaldulensis* had the highest C stock density in the sampled plantation forest, as illustrated in Table 1.

**Table 1.** Summary statistics of the measured C stock plantation forest parameters.

| Statistic (MgCha$^{-1}$) | *Eucalyptus camaldulensis* | | | *Eucalyptus grandis* | | | *Pinus patula* | | |
|---|---|---|---|---|---|---|---|---|---|
| | DBH | Height | C Stock | DBH | Height | C Stock | DBH | Height | C Stock |
| *Mean* | 81.4 | 60.6 | 2485.3 | 67.4 | 70.6 | 405.7 | 56.8 | 58.6 | 377.9 |
| *Median* | 77.4 | 52.7 | 1470.3 | 51.4 | 49.7 | 327.8 | 43.5 | 38.7 | 295.4 |
| *Max* | 231.9 | 88.9 | 8998.2 | 97.9 | 90.1 | 429.8 | 64.3 | 66.6 | 600.3 |
| *Min* | 11.4 | 23.8 | 13.7 | 14.7 | 27.8 | 111.3 | 10.6 | 19.4 | 9.7 |
| *n* | 97 | - | - | 60 | - | - | 34 | - | - |
| *s.td* | 57.6 | - | - | 51.7 | - | - | 48.9 | - | - |

### 3.2. Hierarchical Bayesian Geostatistical Approach

The variogram of residuals of predictors obtained from the Landsat-8 and the Sentinel-2 satellite sensors provided priors of the modelled parameter specifications of $\sigma^2_{\tilde{\varepsilon}}$ and $\sigma^2_w$ for the geostatistical approach, Figure 3a,b, respectively. Sentinel-2-derived predictors have more influence on the spatial distribution of the modelled regionalised variable than what we see with its counterpart from the Landsat-8-derived vegetation indices. The spatial dependence is greatly reduced in the Sentinel-2-derived modelled variogram (Figure 3a,b), as the variable displays a strong spatial structure attributed to the finer spatial and spectral resolutions of Sentinel-2 than that of the Landsat-8 data. The modelled C stock using the Bayesian hierarchical approach, therefore, adopted scale parameters derived from the variogram exploratory analysis of the outcome variable using ancillary data from Landsat-8 and Sentinel-2 [53].

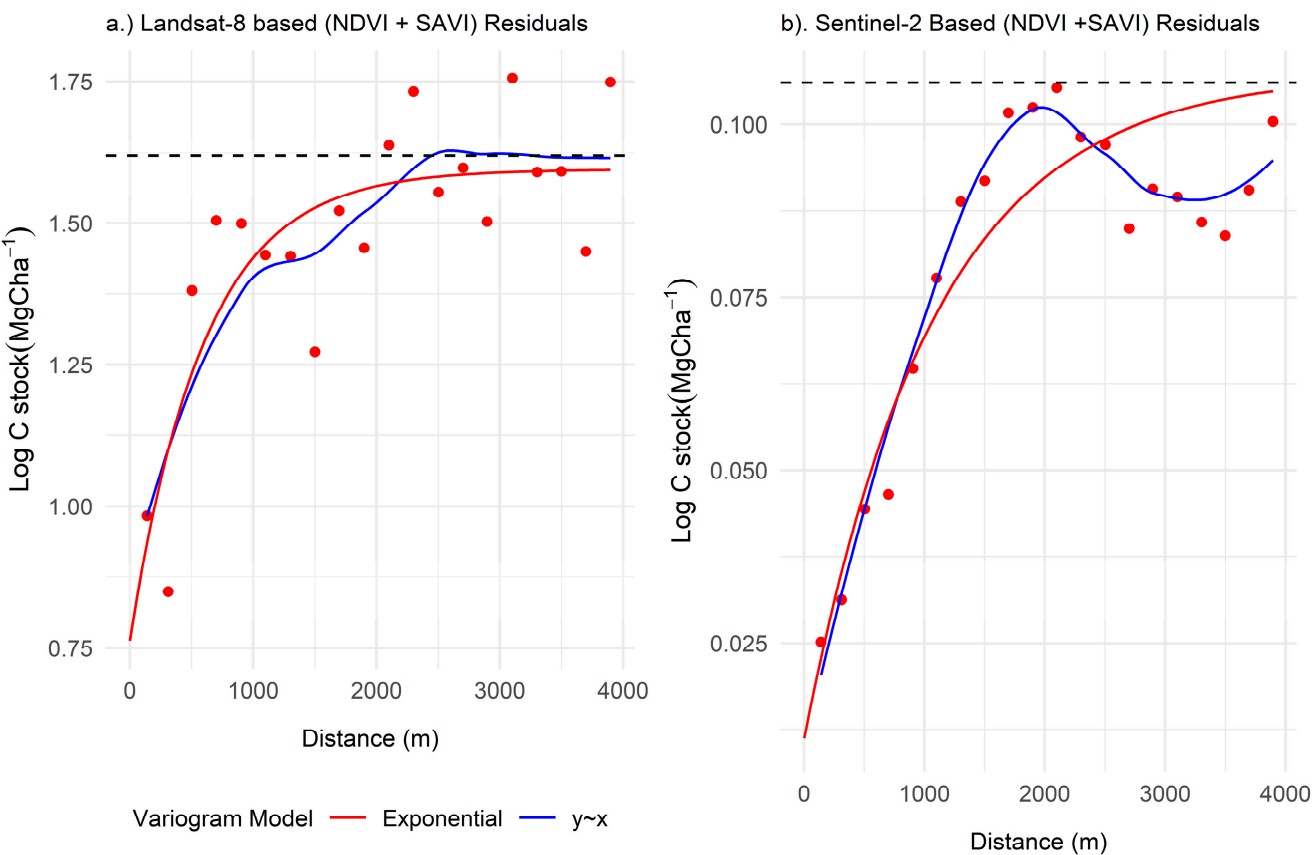

**Figure 3.** Variogram modelling for Landsat-2- and Sentinel-2-derived vegetation indices. The asymptote for the theoretical variogram model is shown by the black dotted line.

In accordance with the foregoing logic, the scale parameter values derived from Figure 3 put on a constraint on the parameter space of the probability distribution function [34]. We fitted the variogram of residuals for both sensors using the exponential covariance model, as illustrated in Figure 3.

### 3.2.1. C Stock and Medium Resolution Sensor-Derived Vegetation Indices

Medium resolution-derived vegetation indices utilised as predictors for C stock modelling in the form of $NDVI$, $SAVI$, and $EVI$ showed different results for the two sensors. The Landsat-8-based C stock model demonstrated $NDVI$ as the only predictor for C stock, whilst the Sentinel-based C stock employing the same covariates illustrated both $NDVI$ and $SAVI$ as significant predictors of the C stock. Amongst the tested Landsat-8- and Sentinel-2-derived vegetation indices, only $NDVI$ was significantly different from zero, as 95% Credible Intervals (CI) for both sensors exclude zero, as illustrated in Table 2.

The 95% CI of $2.63 \leq NDVI \leq 6.30$ and $6.06 \leq NDVI \leq 6.51$ for the Landsat-8- and Sentinel-2-based C stock prediction models, respectively, puts the Sentinel-2-based predictive model in a stronger position due to the strength of the predictor coefficient. It is evident from Table 2 that the posterior distribution of independent variable coefficients for Landsat-8 and Sentinel-2 displayed marked differences. The larger $NDVI$ coefficients in the Sentinel-2-derived C stock model signify the relative preference of using Sentinel-2 for C stock prediction over Landsat-8-derived vegetation indices.

The Landsat-8-derived C stock model has a slightly stronger spatial correlation compared to that of its Sentinel-2 C stock counterpart, as evidenced by the estimates of the effective ranges of the models. We observed an effective range ($\approx \frac{3}{\phi}$) of 2500 m with a ($2142 \leq \phi \leq 2307$) 95% CI for the Landsat-8 C stock, whilst the Sentinel-2-based C stock model gave an effective range of 1667 m with a ($1304 \leq \phi \leq 2142$) 95% CI, as illustrated in Table 2. The Sentinel-2-based $NDVI$ takes away most of the spatial correlation structure in C stock than the same predictor derived from the Landsat-8 OLI does. The differences in the spectral properties between Landsat-8 (11 spectral bands) and Sentinel-2

(13 spectral bands) vindicates this observation [64,65]. In both models of Landsat-8 and Sentinel-2, the spatially structured variance, $\sigma_w^2$, is higher than the white noise variance is [66,67].

**Table 2.** Landsat-8- and Sentinel-2-derived predictors of C stock. NDVI =Normalised Difference Vegetation Index; SAVI = Soil-Adjusted Vegetation Index; EVI = Enhanced Vegetation Index; $\sigma_w^2$ = spatially structured variance; $\sigma_\varepsilon^2$ = White noise; $\phi$ = spatial decay parameter.

| Parameter | Landsat-8 OLI C Stock Model | | | | Sentinel-2 MSI C Stock Model | | | |
|---|---|---|---|---|---|---|---|---|
| | Mean | s.d | 2.5% | 97.5% | Mean | s.d | 2.5% | 97.5% |
| *Intercept* | 1.34 | 0.49 | 0.37 | 2.27 | 0.93 | 0.24 | 1.42 | −0.49 |
| *NDVI* | 4.49 | 0.94 | 2.63 | 6.30 | 6.30 | 0.11 | 6.06 | 6.51 |
| *SAVI* | −0.50 | 0.72 | −1.55 | 1.26 | 0.02 | 0.38 | −0.72 | 0.77 |
| *EVI* | −0.50 | 0.55 | −1.65 | 0.53 | 0.01 | 0.11 | −0.19 | 0.22 |
| $\sigma_w^2$ | 1.47 | 0.39 | 0.76 | 2.22 | 0.07 | 0.01 | 0.053 | 0.10 |
| $\sigma_\varepsilon^2$ | 0.39 | 0.15 | 0.13 | 0.68 | 0.005 | 0.004 | 0.0005 | 0.01 |
| $\phi$ | 0.0013 | 0.000 | 0.0013 | 0.0014 | 0.0012 | 0.0003 | 0.0014 | 0.0023 |

### 3.2.2. Bayesian-Based C Stock Predictions

We fitted models with Landsat-8- and Sentinel-2-derived spectral auxiliary variables for predicting the C stock at unsampled sites within the studied region. Covariates in the form of $NDVI$, $SAVI$, and $EVI$ were derived from a 10,000 m$^2$ gridded raster, thereby making the predicted C stock represent the average values in every raster pixel. $NDVI$ is the only significant predictor for C stock prediction in managed plantation forest ecosystems, as the 95% CIs of the other covariates contain zero (Table 2). The significance of $NDVI$ as a vegetation index correlated with the biophysical properties of vegetation, leaf area index being one of them, is well established [68]. Sentinel-2-derived C stock predictions are more credible than their Landsat-8 derived C stock predictions are. Landsat-8-based C stock predictions have higher uncertainty compared to that of Sentinel-2-based C stock predictions.

This is demonstrated in the 95% posterior predictions illustrated in Figure 4a, indicating the C stock 95% CI to be greater than the Sentinel-2 predicted values are. This makes Landsat-8-based predictions highly uncertain and less precise than the Sentinel-2-based C stock predictions are. Landsat-8 and Sentinel-2-based C stock predictions alongside their 95% CIs are illustrated in Figure 4a,b and Figure 5a,b, respectively. The C stock predictions range between 1 MgCha$^{-1}$ and 497 MgCha$^{-1}$ and between 1 MgCha$^{-1}$ and 285 MgCha$^{-1}$ for the Landsat-8 and Sentinel-2 sensors, respectively. Higher C stock values are predicted in the southern part of the studied region in the Landsat-8-based predictive models than those in the Sentinel-2 models. In spite of this trend, the Sentinel-2-based C stock predictive model displays higher C stock values uniformly across the study area with a smaller magnitude than the Landsat-8-based models do (Figure 4a,b and Figure 5a,b).

As such, the Sentinel-2 model seems to underpredict more of the C stock at unsampled locations compared to that of its Landsat-8-based C stock predictive model counterpart. The slight underprediction by Sentinel-2 can partly be attributed to the finer spatial and spectral resolutions of the sensor within the visible and near-infrared segments of the electromagnetic spectrum (EMS) [16,69,70]. Enhancements in the spectral and spatial resoultion of Sentinel-2 confirms the much shorter 95% Credible Interval Widths (CIWs) displayed by the Sentinel-2-based C stock predictive model in Figure 4d (0.40–1.78 MgCha$^{-1}$) than those of the Landsat-8-based predictive model in Figure 4b (2.0–4.7 MgCha$^{-1}$). Predicted values in both new generation remote sensing-based models look better compared to the ones reported in previous research including Jiang et al. The authors of [71] established a mean AGB RMSE of 40.9 MgCha$^{-1}$ in north-east China, and the authors of [72] determined an AGB RMSE of 36.67 MgCha$^{-1}$ in Vietnam using the Random Forest (RF) algorithm.

Furthermore, Takagi et al. [73] employed LiDAR for the prediction of forest biomass in Hokkaido, Japan, and determined an RMSE biomass prediction of 19.1 MgCha$^{-1}$. The differences between the prediction accuracy results reported in the literature and our study can also be justified by the differences in forest density, since the erstwhile studies were carried out in subtropical rainforest biomes.

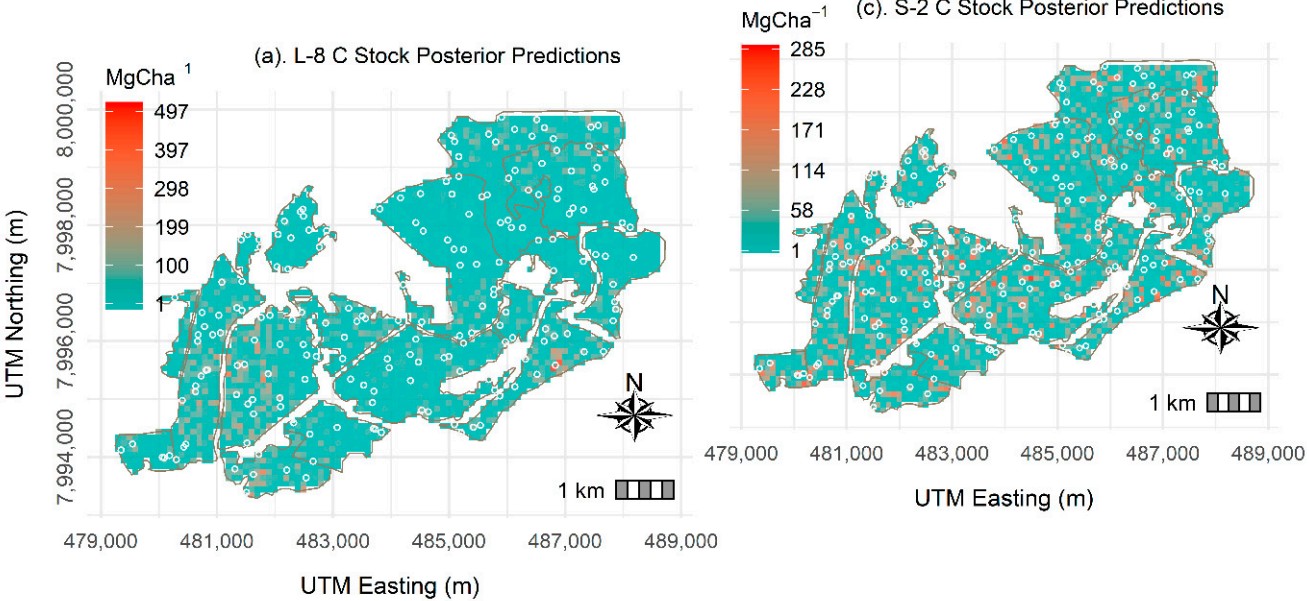

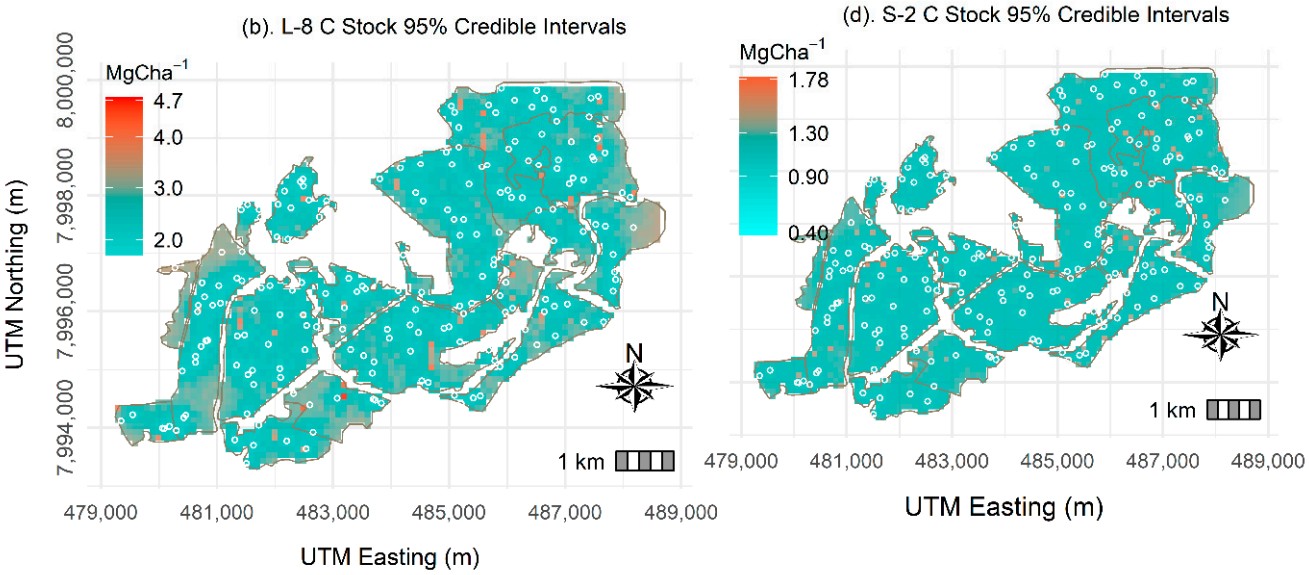

**Figure 4.** Bayesian Landsat-8 (L-8)- and Sentinel-2 (S-2)-based C stock posterior predictions alongside 95% credible intervals.

### 3.2.3. Model Validation and Diagnostics

The *k-fold* cross-validation metrics employed in the assessment of the Bayesian-based predictive models are presented in Table 3. The Sentinel-2-based C stock predictive model is the top performing model in terms of the RMSE (0.17 MgCha$^{-1}$) and Mean Absolute Error (MAE) (0.13 MgCha$^{-1}$). The Sentinel-2 model presents a predictive ability that is almost at the benchmark nominal coverage of 95% [60,73]. The Landsat-8-based C stock predictive model has an 85.4% coverage for the 95% prediction intervals coupled with higher RMSE and MAE values (Table 3).

Figure 5a,b illustrates the scatterplots of observed C stock against the predicted C stock, alongside the 95% intervals for both Landsat-8 and Sentinel-2 C stock-based predictive models. Evidence of the Sentinel-2-based C stock predictive model performing better than its Landsat-8 C stock-based counterpart did is clear from the scatter plot of the model in Figure 5b. It is evident from the model

diagnostics illustrated in Figure 5a that the Landsat-8-based C stock predictive model tends to over-predict more C stock values compared to those of the Sentinel-2 C stock-based predictive model (Figure 5b). This makes the Sentinel-2 C stock-based model favourable compared to the Landsat-8 predictive model.

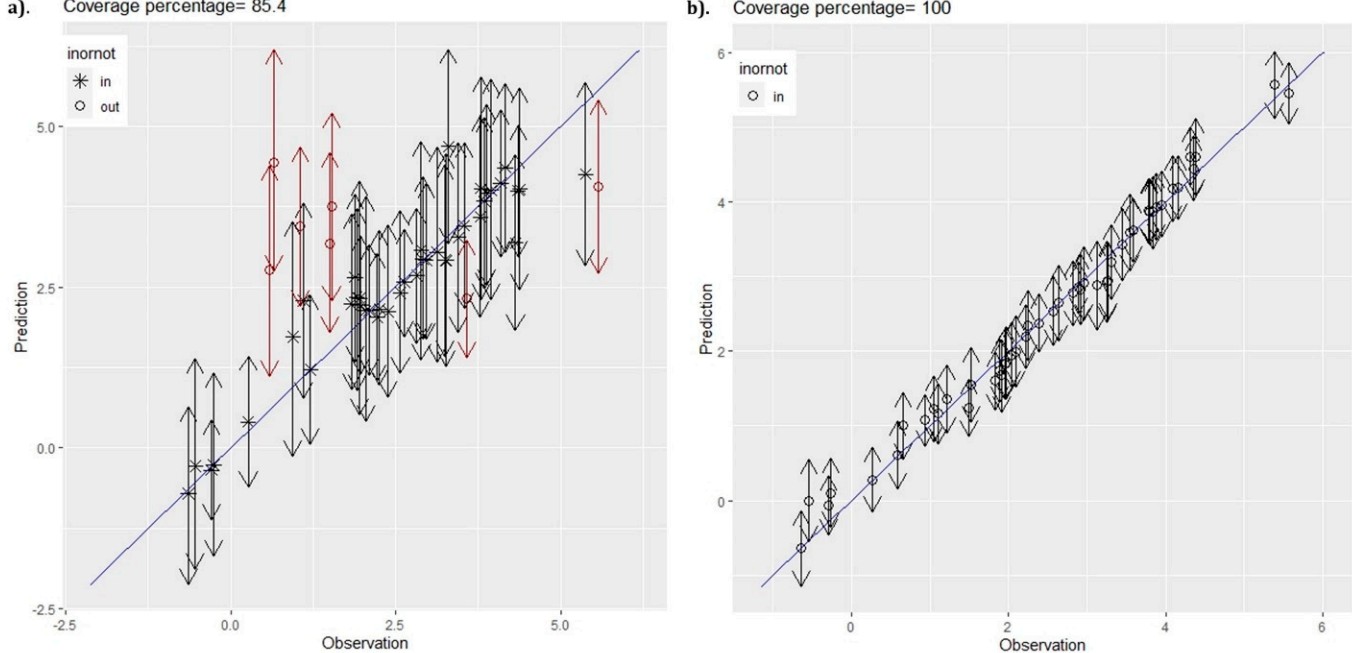

**Figure 5.** (**a**) Predictions (MgC ha$^{-1}$) against observed C stock (MgCha$^{-1}$) for the Landsat-8-based spatial model. (**b**) Predictions against observed C stock for the Sentinel-2-based spatial model alongside 95% intervals.

**Table 3.** Validation statistics for C stock Bayesian-based C stock predictive models.

| Model Evaluation Criterion | Landsat-8-Derived Predictors | | | Sentinel-2-Derived Predictors | | |
|---|---|---|---|---|---|---|
| | Independent Error Model | Spatial Intercept Only Model | Spatial Model | Independent Error Model | Spatial Intercept Only Model | Spatial Model |
| *RMSE* (Mgha$^{-1}$) | 1.23 | 0.97 | 0.97 | 0.31 | 1.18 | 0.17 |
| *MAE* (Mgha$^{-1}$) | 0.93 | 0.53 | 0.57 | 0.26 | 0.77 | 0.13 |
| *CRPS* (Mgha$^{-1}$) | 0.72 | 0.38 | 0.38 | 0.20 | 0.56 | 0.14 |
| *CVG* (%) | 91.67 | 85.42 | 85.42 | 95.83 | 89.58 | 100.00 |
| *DIC* | 220.8 | 48.40 | 71.0 | −201.3 | 283.5 | −564.5 |

### 3.3. Frequentist Geostatistical Modelling

#### 3.3.1. C Stock Density

We assessed the C stock density of the studied region using species type as a possible source of C stock density variability and established the mean C stock of *Eucalyptus camaldulensis*, *Eucalyptus grandis*, and *Pinus patula* species to be 35.10 MgCha$^{-1}$, 37.38 MgCha$^{-1}$, and 29.45 MgCha$^{-1}$, respectively. Despite *Pinus patula* being the most dominant species in the sampled region (Figure 1), Eucalyptus *grandis* has the highest concentration of C stock. An evaluation of the C stock density of the various plantation forest species making up the study area using Analysis of Variance (ANOVA) shows that the C stock densities of the different tree species are not significantly different from each other ($F_{2188} = 0.21$, $p = 0.811$). Tukey–Kramer [74] multiple comparison test was conducted in order to avoid the risk of accumulating false positives as a result of multiple tests being carried out at the same time, and the results, also not significantly different. An evaluation of C stock density categorised

by management style conducted in Nepal using frequentist ordinary kriging and KED geostatistical methodologies demonstrated significant differences between the C stock densities stored by different community forests [75].

However, the present study evaluated the prediction performance of geostatistical models using probabilistic- and likelihood-based frameworks, without stratifying the prediction performances by forest species as was performed by the authors of [75]. This would be an interesting research question for the future, as it is evident that different forest species of tropical and subtropical biomes store significantly different biomass and C stocks, an issue that forest practitioners would be interested in knowing for their investment options and decisions [9].

### 3.3.2. Landsat-8- and Sentinel-2-Based C Stock Linear Modelling

The linear modelling of C stock using sampled C stock data and multispectral remotely sensed data showed $NDVI$ to be a significant predictor for the Landsat-8-derived vegetation indices, whilst $NDVI$ and $SAVI$ were significant predictors for the Sentinel-2-derived vegetation indices (Table 4).

**Table 4.** Landsat-8 and Sentinel-2 best linear models of feature space.

| Predictors | Landsat-8-Based Linear Model | | | Sentinel-2-Based Linear Model | | |
|---|---|---|---|---|---|---|
| | Coefficient | *p*-Value | *α* = 0.05 | Coefficient | *p*-Value | *α* = 0.05 |
| *Intercept* | 0.03 | 0.93 | *Insignificant* | −0.30 | 0.31 | *Insignificant* |
| *NDVI* | 7.67 | 0.00 | *Significant* | 6.69 | 0.00 | *Significant* |
| *SAVI* | 1.04 | 0.11 | *Insignificant* | 1.24 | 0.01 | *Significant* |
| *EVI* | 0.33 | 0.57 | *Insignificant* | 0.20 | 0.15 | *Insignificant* |

### 3.3.3. Landsat-8 and Sentinel-2-Based KED Predictions

Figure 6a illustrates the Landsat-8-based C stock KED predictions alongside their prediction variances. The results of KED using the spatial dependencies in the outcome variable gave C stock predictions with a range of $1 \leq \text{MgCha}^{-1} \leq 327$ and a corresponding standard error range of $1.1 \leq \text{MgCha}^{-1} \leq 5.1$. As shown in Figure 6b, C stock predictions similar to the sampled data displayed less uncertainty compared to that of the predictions made at remote locations within the study domain. Thus, the Landsat-8-based KED C stock variances illustrated in Figure 6b are not better than 26.01 (=5.1$^2$) $\text{MgCha}^{-1}$ for the studied managed plantation forest ecosystem. A high C stock density dominates the predictions at the original support (500 m$^2$), where primary data were derived using sampled field data. This is because kriging is a geostatistical method that characterises the values of an outcome variable (C stock) similar to the original data locations, which tend to have more similar statistical properties to the sampled value at that point than those of the values obtained in remote locations [75,76].

As illustrated in Table 4, we made Sentinel-2-based C stock KED predictions using $NDVI$ and $SAVI$ as aides of the predictive model, and these were made after ordinary variogram modelling of the primary variable. Incorporation of $NDVI$ and $SAVI$ as independent variables gave a reduced total sill of the modelled variogram (Figure 3b) and the subsequent shortening of the range of spatial dependence. Hence, Sentinel-2-derived vegetation indices utilised as predictors in the C stock model predicted C stock with a $1 \leq \text{MgCha}^{-1} \leq 290$ range and an accompanying standard error ranging from $2 \leq \text{MgCha}^{-1} \leq 11$ (Figure 6c,d). Finer spatial and spectral characteristics within the visible and NIR of the Sentinel-2 satellite sensor resulted in $SAVI$ being incorporated in the C stock predictive model, in addition to $NDVI$ [77,78].

Locations closer to the margins of the sampled domain show an increasing trend in the prediction standard error, as these locations are farther away from the sampled C stock observations. As Figures 3b and 6d show, the significant reduction of the range of spatial dependence in the modelled C stock data demonstrate how the Sentinel-2-driven model of feature space carry both $NDVI$ and $SAVI$ in the spatial correlation structure of the outcome variable. In line with theory, the KED calculated error variance appears to rely on the sampled data configuration, in which uncertainty decays towards the sampling sites [66,79].

As the results in Figure 6a–d show, the Landsat-8-based C stock model predicts more C stock in the southern and northern parts of the sampled region of the sampled domain. On the contrary, the Sentinel-2-based C stock predictive model predicts more C stock uniformly across the sampling domain. This can be explained by the precision with which the much-improved spatial resolution of

Sentinel-2 allows the sensor's sensitivity to the forest parameter spectral signal within the NIR and visible regions of the electromagnetic spectrum.

### 3.3.4. Frequentist Geostatistical Predictive Model Evaluation

We evaluated the frequentist-based C stock predictive models using leave-one-out cross-validation statistics and model residual diagnostics. As illustrated in Table 5, the Sentinel-2-based C stock predictive model has the lowest RMSE compared to that of the Landsat-8-based C stock model. The validation statistics shown in Table 5, therefore, suggest the Sentinel-2-based KED C stock model to be the most ideal under the frequentist geostatistical approach, as it has the best characteristics. This is evident from the predictive model's RMSE (1.19 MgCha$^{-1}$) and PRESS (6.06).

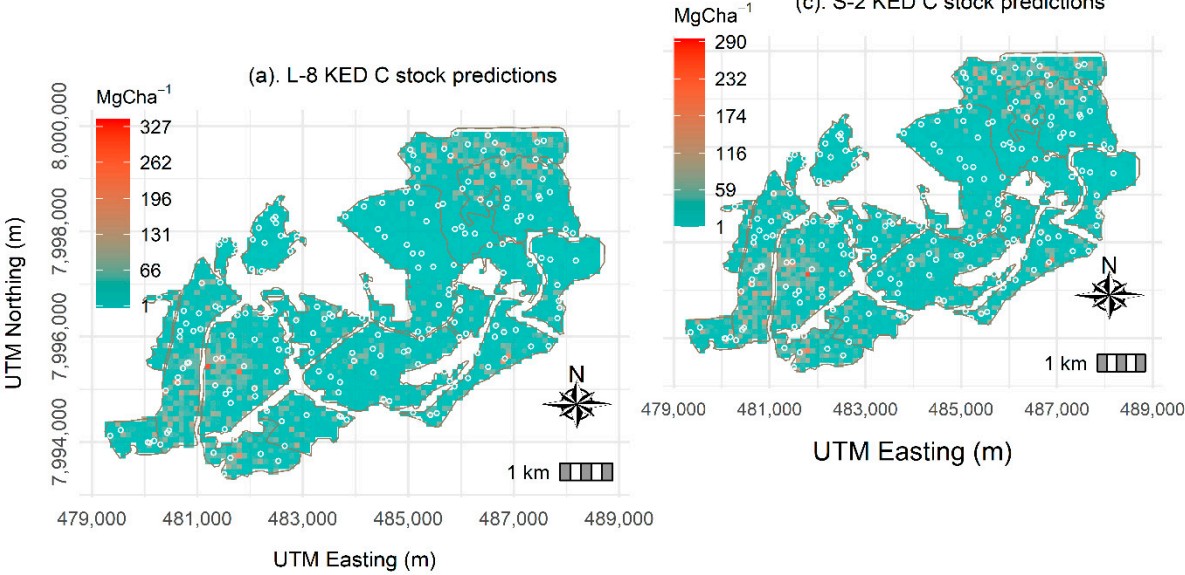

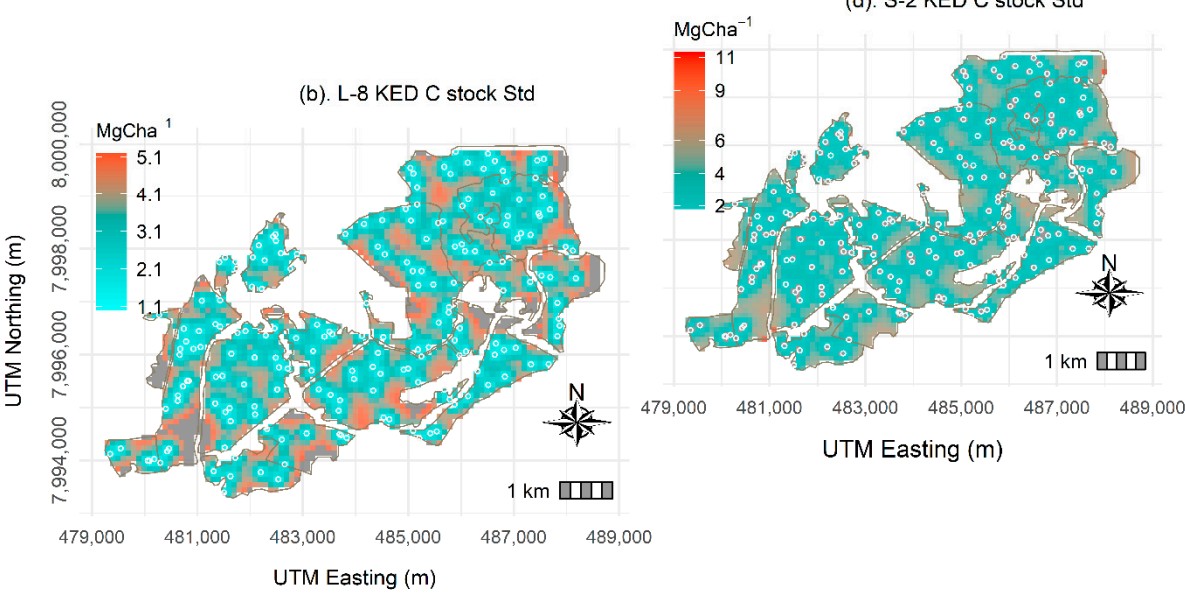

**Figure 6.** Frequentist Landsat-8- (L-8) and Sentinel-2 (S-2)-based C stock predictions and 95% confidence intervals.

The supremacy of the Sentinel-2-based C stock predictive model is also corroborated by the model diagnostics illustrated in Figure 7, showing robust model residuals that are evidently insensitive to outlying observations. Furthermore, the test statistics regarding the normality of the model residuals for the two kriging variants (Table 6) show the $p$-values of more than the 0.05 test statistic,

leading to the rejection of the null hypothesis of non-normal residual errors. The rejection of the Shapiro normality null hypothesis is necessary for both Landsat-8 and Sentinel-2-based C stock predictive models (Table 6). The best linear model of feature space making use of Landsat-8- and Sentinel-2-derived vegetation indices gives the Sentinel-2-based C stock predictive model a more symmetrical error distribution than that of its Landsat-8-based counterpart (Figure 7). This fact is confirmed and reinforced by the residuals of the Sentinel-2 C stock predictive model.

**Table 5.** Frequentist geostatistical C stock prediction validation statistics.

| Predictors | Landsat-8-Based C Stock Predictions | | Sentinel-2-Based C Stock Predictions | |
| --- | --- | --- | --- | --- |
| | Ordinary Kriging (OK) | Kriging with External Drift (KED) | Ordinary Kriging (OK) | Kriging with External Drift (KED) |
| *ME* | 1.01 | 1.00 | 1.01 | 1.00 |
| *MSE* | 1.01 | 1.00 | 1.01 | 1.00 |
| *RMSE* | 2.94 | 2.84 | 2.91 | 1.19 |
| *PRESS* | 222.34 | 208.42 | 222.34 | 6.06 |

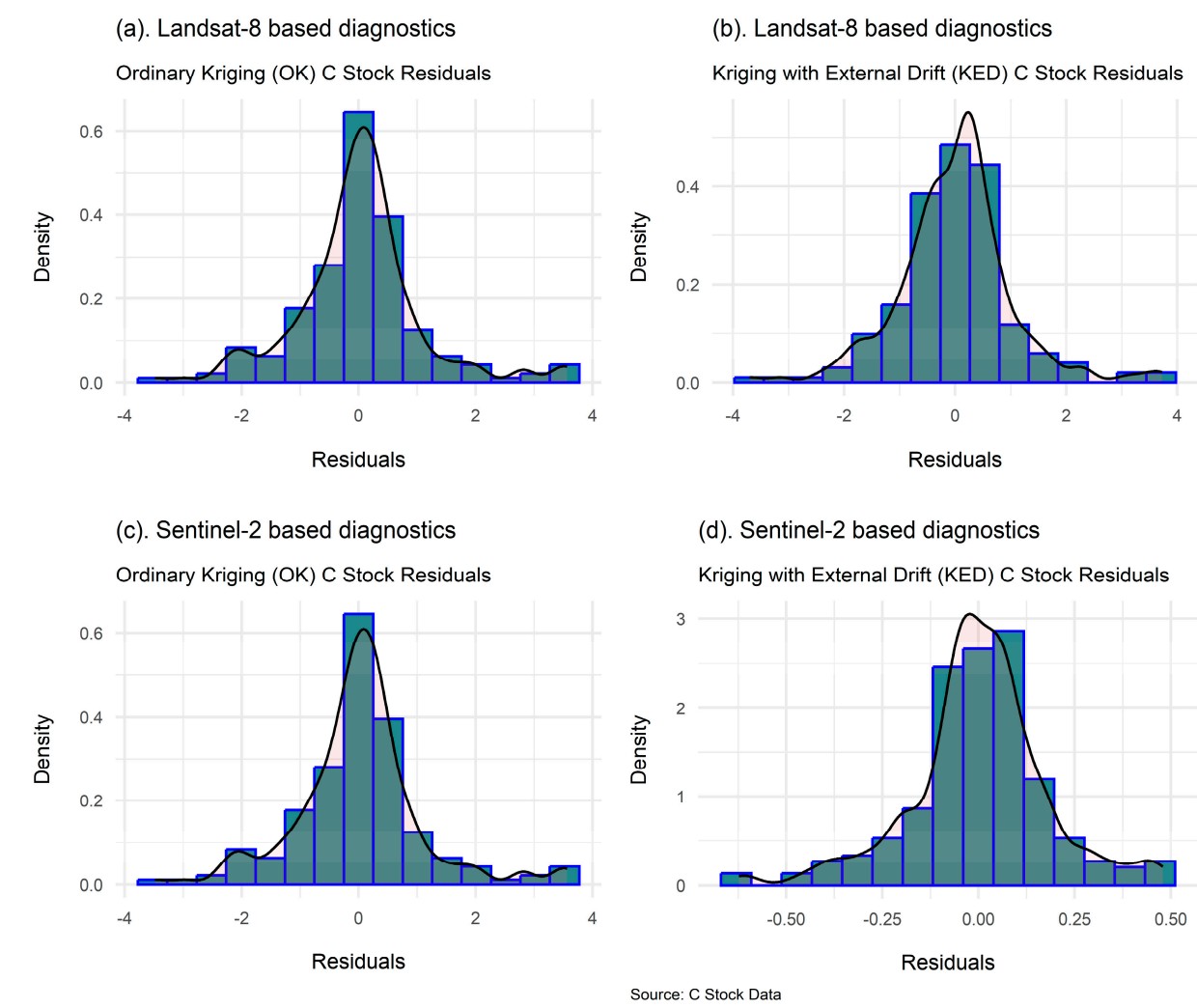

**Figure 7.** Frequentist-based Landsat-8 and Sentinel-2 C stock KED residual diagnostics.

**Table 6.** Frequentist geostatistical model diagnostics test statistics.

| Modelling Approach | Test Statistic | *p*-Value | Modelling Technique |
|---|---|---|---|
| Frequentist approach | 0.097 | 0.264 | Landsat-8 |
| Frequentist approach | 0.132 | 0.136 | Landsat-8 |
| Hierarchical Bayesian approach | 0.975 | 0.367 | Sentinel-2 |
| Hierarchical Bayesian approach | 0.773 | 0.278 | Sentinel-2 |

*3.4. Bayesian- and Frequentist-Based C Stock Predictive Model Summaries*

A summarised overview of the performances of the tested inferential approaches, one using the Bayesian framework and the other one using the frequentist approach, are illustrated in Table 7. We deduced from the summary statistics depicted in Table 7 that all the Landsat-8-based C stock predictive models from either the Bayesian or the frequentist paradigms tend to overpredict the modelled C stock values, whilst the Sentinel-2-based C stock-based predictive models gave C stock values that are just within the range. The Sentinel-2-based C stock predictive model from the Bayesian statistical paradigm offers the best qualities of C stock prediction model in terms of the quality of predictions and ability to predict true values outside the sampled data (Table 7).

**Table 7.** Summaries of the Bayesian and the frequentist geostatistical approaches. *CIWs* = Confidence Interval Widths; ME = Mean Error; RMSE = Root Mean Square Error.

| Validation Criterion | Bayesian Geostatistical Approach | | Frequentist Geostatistical Approach | |
|---|---|---|---|---|
| | Landsat-8-Based C Stock Model | Sentinel-2-Based C Stock Model | Landsat-8-Based C Stock Model | Sentinel-2-Based C Stock Model |
| **RMSE** | 0.97 | 0.17 | 2.84 | 1.19 |
| **ME** | 0.57 | 0.13 | 1.01 | 1.00 |
| **Error/CIWs** | $2 \leq MgC \leq 4.7$ | $0.4 \leq MgC \leq 1.8$ | $1.1 \leq MgC \leq 5.1$ | $2 \leq MgC \leq 11$ |
| **Prediction range** | $1 \leq MgC \leq 497$ | $1 \leq MgC \leq 285$ | $1 \leq MgC \leq 327$ | $1 \leq MgC \leq 290$ |
| **Conclusion** | Overprediction | Perfect | Overprediction | Perfect |

## 4. Discussion

*4.1. Bayesian Geostatistical Approach and C Stock Predictions*

The hierarchical design of stochastic models is intrinsically related to Bayesian inference, where the probability distribution of observed C stock (in the present case) is specified in a hierarchical fashion [61]. From the three tested predictors of $NDVI$, $SAVI$, and $EVI$ for C stock prediction, we established $NDVI$ as the only significant covariate of the modelled C stock variable for both the Landsat-8- and Sentinel-2-derived C stock predictive models. Because of the higher spatial resolution of Sentinel-2, $NDVI$ emerged as a stronger predictor of C stock for the Sentinel-2-based model than it was for the Landsat-8-based C stock predictive model [78].

The importance of $NDVI$ as a vegetation index correlated with the biophysical properties of vegetation such as the Leaf Area Index (LAI), is well known [68]. This shows the relative preference of employing Sentinel-2 for C stock prediction over Landsat-8 derived vegetation indices. Consequently, Sentinel-2 derived C stock predictions have better credibility than the Landsat-8-derived equivalent does. Landsat-8-based C stock predictions have more uncertainty compared to that of the Sentinel-2-based C stock predictions. The utilisation of new generation remote sensing-derived vegetation indices as predictors of C stock for a managed plantation forest ecosystem under a Bayesian framework is unique [80]. Previous studies, including Babcock et al. [14] and Babcock et al. [81], have used LiDAR on its own to obtain predictors of forest biomass rather than a comparative approach such as that we employed in this research.

The tendency of underprediction in Sentinel-2-derived C stock models can partly be attributed to the finer spatial resolutions of the sensor within the visible and near-infrared segments of the electromagnetic spectrum (EMS) [16,69,70]. As highlighted in [78], there are notable improvements in the radiometric resolution of the Landsat-8 OLI sensor from 8 bits to 12 bits.

Enhancements in the spatial resolution of Sentinel-2 confirms the much smaller 95% Credible Interval Widths (*CIWs*) displayed by the Sentinel-2-based C stock predictive model. The predicted values in both new generation remote sensing-based models look more attractive compared to the ones reported in previous research, including that of Jiang et al. [71], who established a mean AGB RMSE of 40.9 MgCha$^{-1}$ in north-east China, and that of Dang et al. [72], who determined an AGB RMSE of 36.67 MgCha$^{-1}$ in Vietnam using the Random Forest (RF) algorithm. Furthermore, Takagi et al. [73] employed LiDAR for the prediction of forest biomass in Hokkaido, Japan, and determined an RMSE biomass prediction of 19.1 MgCha$^{-1}$. Differences between the prediction accuracy results reported in the literature and our study can also be justified by the differences in forest densities, since the erstwhile studies were carried out in subtropical rainforest biomes [70–72]. For instance, Japan is regarded as the most forested country in the world, with approximately 70% of its land being forested, which is more than that of the location of the present study, Zimbabwe, where about 40% or 15,624,000 ha of the land is forested [82].

Machine learning methods for the mapping of AGB premised on Landsat-8 imagery were compared by Wu et al. [83] and Xiong and Wang [84], and a Random forest method with an RMSE of 26.43 tons/ha was found to be superior to the other methods such as stochastic gradient boosting, k-nearest neighbour, and support vector regression. The current study establishes AGB accuracies using the Bayesian and the frequentist geostatistical approach with the prediction accuracies of the former one outweighing those of the latter one. The Bayesian methods utilised in this study are both superior to both the frequentist approach and methods utilised in the literature for the mapping of AGB. However, Bayesian methods are not easy to implement and adopt by ordinary forest practitioner due to their complexity [51]. It is this complexity and the lack of simple software packages that greatly hinder their adoption and operationalization for forestry monitoring.

### 4.2. Frequentist Geostatistical Approach and C Stock Predictions

The geostatistical kriging variant, KED, provides an optimal geostatistical technique under the frequentist paradigm that is employed in the description of spatial patterns and predicting values of a variable at unsampled locations and, consequently, it evaluates the uncertainty associated with the predicted values [19]. We employed and subjected the C stock model to the same new generation remote sensing-derived predictors of vegetation indices as we did under the Bayesian approach and established some notable differences and similarities. Similar to the Bayesian C stock model for Landsat-8, *NDVI* also presented as the only significant predictor in the frequentist Landsat-8-based C stock predictive data. However, both *SAVI* and *NDVI* were significant predictors for the frequentist Sentinel-2-based C stock model. We employed KED for both the Landsat-8- and the Sentinel-2-based C stock models. The finer spatial scale of the Sentinel-2 sensor, coupled with the wider spectrum of its NIR band (760–900 nm) compared to the narrow spectrum of Landsat-8 NIR band (850–880 nm), resulted in *SAVI* and *NDVI* being incorporated in the C stock predictive model [77,78]. The NIR band is widely known to be vital to the biophysical factors of vegetation assessment and monitoring. Consequently, the spectral response profile of Landsat-8 and Sentinel-2 display some minor differences in the NIR and visible regions of the electromagnetic spectrum, which could explain differences in the prediction performances, as established in the current study [62,81].

KED has been widely applied for the estimation of AGB using other factors as predictors of the forest biomass [18,76]. For instance, Korhonen et al. [20] made an average AGB prediction of 32 Mgha$^{-1}$ in the Brazilian Amazon. KED performed better in this study than the pure-based approach of the ordinary kriging algorithm did, as *NDVI* and *SAVI* formed the best linear model of feature space in the Sentinel-2 C stock predictive model. Most studies comparing the prediction performances of Landsat-8 and Sentinel-2, including those by Korhonen et al. [23] and Meyer et al. [85], have not established significant or systematic differences in the predictive performance of the aforementioned sensors. In the current study, both Bayesian and frequentist Landsat-8-based C models tend to overpredict the modelled and estimated C stock values across the sampled domain, as their range falls outside the range of observed data. This implies that forest practitioners can feasibly exploit the scale and band spectrum characteristics of Sentinel-2 for representative and accurate C reporting necessary for monitoring and verifying climate change mitigation actions.

### 4.3. Comparative Bayesian and Frequentist C Stock Predictive Model Evaluation

Sentinel-2-based prediction of C stock using the frequentist-based KED illustrates predictions within the range of measured C stock values, but the variances are high compared to those of the same predictions made using Sentinel-2 within a Bayesian inferential framework. On the contrary, both Bayesian and frequentist C stock predictive models constructed using Landsat-8 overpredicted

the sampled C stock, as the range of predicted values fell outside the observed C stock values. This observation is further bolstered by the results of the diagnostic residuals of the models from the frequentist and the Bayesian techniques, where the plots of the observed versus the predicted values of C stock are perfectly predicted in the Bayesian-based Sentinel-2 models. As reported in the study, Sentinel-2-based C stock predictive models are more accurate than their Landsat-8 equivalents are for both the Bayesian and the frequentist inferential approaches. Previous studies mapping AGB accuracies using either the frequentist or the Bayesian approaches, coupled with satellite imagery data of Landsat-8 and Sentinel-2, report lower prediction accuracies [22,71]. As hypothesised, this confirms the superiority of the Bayesian geostatistical approach in handling the uncertainty and improving the accuracy for predicting C stock over the frequentist geostatistical approach.

Important research in the field of biomass estimation using the Bayesian techniques include [13–16]. Recent studies assessing AGB distribution in temperate European ecosystems established accuracies of 17.52 Mg/ha by Babcock et al. [14], and 1.16 MgCha$^{-1}$ and 2.69 MgCha$^{-1}$ accuracies were found for Sentinel-2-based and Landsat-8-based C stock predictive models, respectively [16]. Other remote sensing- and machine learning-based efforts for the estimation and prediction of AGB in recent times include those by Do et al. [17], who found that mangrove AGB predictions in Vietnam range from 6.51 to 368 Mgha$^{-1}$ and from 13.70 to 320.1 Mgha$^{-1}$ for remote sensing and Artificial Neural Networks, respectively. Because the present study utilised data from improved remote sensing platforms for predicting C stock, our reported accuracies surpass those reported in the literature utilising different methodologies with the same satellite sensors with lower quality.

Past studies utilising the frequentist geostatistical approach separate from the Bayesian technique for C stock estimation are also well documented in the literature. The authors of [20] mapped AGB in the Brazilian Amazon using Kriging with External Drift and established prediction accuracies for different sample sizes ranging from 0 to 110 km for distances within 300 km radii from the prediction locations. The lowest RMSE for the estimated AGB for a sample size of 110 was 32.8 Mgha$^{-1}$, whilst the lowest accuracy for the lowest sample size of $n > 0$ was 48.06 Mgha$^{-1}$. Furthermore, Jiang et al. [21] predicted AGB in the Wangyedia forest farm in China using Landsat-8 and the newly launched Landsat-9 and reported RMSEs of 16.83 tha$^{-1}$ and 17.91 tha$^{-1}$, respectively. The authors of [22] coupled remote sensing-derived Sentinel-2 explanatory variables with geostatistics and machine learning algorithms in Mayanmar for predicting the aboveground biomass and established accuracies of 24.91 Mgha$^{-1}$ and 34.72 Mgha$^{-1}$ for the Random Forest-based ordinary kriging and the Random Forest-based co-kriging, respectively.

As demonstrated by the results in [21], Landsat-8 built AGB estimation models can still be superior to AGB models built using the successor Landsat-9 sensor, despite the relative spectral and radiometric improvements in the latest Landsat-9 sensor. This is also justified by the results of the present study as the results from both the Landsat-8 and Sentinel-2-based Bayesian and frequentist approaches are not significantly different from each other. From a practitioner's point of view, the recommendation for the use and adoption of a particular sensor and statistical approach largely depends on the policy problem being addressed. Furthermore, the costs involved and ease of use and application of the sensors and methodology also play a bigger role. Despite its lack of simplicity and difficulties in implementation, the Bayesian approach is more appealing and pragmatic for natural resources monitoring and reporting, as different studies of a particular region would eventually constitute a database that subsequent studies on carbon assessment would rely on for the updating of priors and posteriors in the prediction process.

## 5. Conclusions

This study set out to compare the prediction performances of two inferential geostatistical frameworks, one making use of a hierarchical Bayesian geostatistical approach, and the other one utilizing a frequentist geostatistical approach, for C stock prediction in a managed plantation forest ecosystem in the eastern highlands of Zimbabwe. Broadband spectral indices from two multispectral remote sensing platforms of Landsat-8 and Sentinel-2 were employed as prediction aids for the geostatistical methodologies. We established notable differences between the two inferential approaches in the prediction of C stock, with the Sentinel-2-based hierarchical Bayesian geostatistical approach yielding lower ($0.4 \leq$ MgCha$^{-1} \leq 1.8$) prediction uncertainty values than its frequentist geostatistical KED ($2 \leq$ MgCha$^{-1} \leq 11$) modelling counterpart did. In both geostatistical methods, the Sentinel-2-driven C stock prediction models outperformed the Landsat-8-driven C stock prediction model counterparts.

The bigger policy problem pertaining to climate change mitigation and climate change action being addressed in this study requires accurate and sustainable carbon accounting and verification tools. In that regard, despite the ease of application and use of the frequentist inferential method-

ology, we conclude that the Bayesian-based technique, coupled with high-quality remote sensing information, is a better method for predicting C stock. This is a critical steppingstone in accounting for carbon to achieve climate change adaptation and mitigation under the United Nations Framework Convention on Climate Change (UNFCCC). The Bayesian-based Sentinel-2 C stock predictive model is preferable to its Sentinel-2-based frequentist counterpart for assisting natural resources managers and other forest practitioners in providing advice to governments on decision for afforestation and reforestation.

Using the Bayesian concept that "today's posterior is tomorrow's prior", we are able to construct a meaningful database of forest parameters, specifically C stock parameters, that will aid future estimation and prediction, even with limited datasets, thereby eliminating the need for costly sampling campaigns. The sustainability of such databases is further made better by the availability of freely available satellite data that are continuously being improved in terms of quality and scale of coverage. Conversely, the frequentist geostatistical approach cannot work optimally under limited sample sizes, and therefore, this makes it a costly alternative in the long term for updating C stock databases kept for C stock monitoring and accounting. We therefore recommend and conclude that the Bayesian-based C stock prediction method, coupled with high-quality remote sensing information such as that of Sentinel-2, is a useful inferential statistical methodology for reporting C stock in managed plantation forest ecosystems.

**Author Contributions:** Writing—original draft, T.S.C.; Writing—review & editing, T.D.; Project administration, O.M. All authors have read and agreed to the published version of the manuscript.

**Funding:** This work is based on the research supported in part by the National Research Foundation of South Africa (Grant Number: 84157).

**Data Availability Statement:** Not applicable.

**Acknowledgments:** The authors extend their gratitude to Kutsaranga, Matowanyika and Mukwekwe for giving us access to the plantation forest of the sampled region at Lot 75A of Nyanga Downs in Manicaland Province. This research received support from the University of KwaZuluNatal, college of Agricultural, Earth and Environmental Science.

**Conflicts of Interest:** The authors declare that they have no financial interest nor personal relationships that could otherwise sway the work reported and presented in this article.

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
