# Peer review of "Carbon Stock Prediction in Managed Forest Ecosystems Using Bayesian and Frequentist Geostatistical Techniques and New Generation Remote Sensing Metrics"

_remotesensing, doi:10.3390/rs15061649_

Round 1

Reviewer 1 Report

This article made a prediction of carbon stocks in managed forest ecosystems using Bayesian and frequentist geostatistical techniques and the generation of new remote sensing metrics. The results of this study can shed light on more precise models for estimating carbon in forests. However, there are some corrections that authors should make before publication.

1. A large number of citations are observed throughout the article of a narrative type, where proper use of the style is not being made. The names of the authors are not named properly. It is necessary to correct (for example it says “Least Squares and Bayesian in Bogotá, Colombia, was tested by [12] in a context of hedonic estimation”; it should read “Least Squares and Bayesian in Bogotá, Colombia, was tested by Ghosh & Carriazo [12] in a context of hedonic estimation”)

2. Correct the references. The initials of the authors' names are considered first when the last name should be considered first.

3. Figure 1 does not seem suitable for publication. It is recommended to consider another image arranged in a single box, with a single background format, grid, and coordinates (geographical coordinates). Also, the exact location in the world. The use of GIS software such as Qgis or Arcgis is recommended.

4. The tables must be in the format that the journal requires.

5. It is recommended to standardize the backgrounds of the figures, either in white or with a lead background.

6. Figures 4 and 5. You should consider placing them together to have a better overview of the comparison. Also, use the same unit units as the location map coordinates (geographic coordinates). Same suggestions for figures 8 and 9. These can also be edited in GIS software like Qgis or Arcgis.

7. There are some lines in the results that should be discussed, due to the fact of comparing the results with other investigations (for example lines 495 - 498).

8. It is suggested to place table 6 in the results part.

Author Response

Thank you for the comments and suggestions. Please find attached document for our responses to the comments and suggestions made.

Reviewer 2 Report

The manuscript  titled Carbon stock prediction in managed forest ecosystems using Bayesian and frequentist geostatistical techniques and new generation remote sensing metrics by Chinembiri et al. presents findings of a study comparing traditional versus Bayesian approaches to predict forest aboveground biomass using remote sensing derived metrics. The authors use linear modelling and geostatical kriging methods and use spectral indices derived from two multispectral remote sensing data, Landsat and Sentinel to achieve their objectives. The findings of the study are relevant, important and improve our understanding of geostatistical approaches for aboveground biomass estimation in the field of forest remote sensing and therefore is worthy of publication in the journal Remote Sensing.

However, there are some major improvements that need to be made and issues that need to be addressed in the manuscript before it can be ready for publication. In particular, I had difficulty understanding and following some parts of the Methods and Results. More clarification and elaboration and updating of some of the figures is needed. There’s identical repetition and some odd syntax and structures in the Results and Discussion some of which I indicated. I listed my detailed comments, critiques and suggestions in the attached pdf document.

 I look forward to seeing the revised version. 

Author Response

Thank you for the comments and suggestions. Please find attached the document for our responses to the comments and suggestions.

Reviewer 3 Report

This could be a valuable study, but I think that the authors may have overemphasised the importance of the statistical framework employed. While it is commendable to demonstrate an alternative framework, i.e. Bayesian, for the estimation of carbon stocks,the authors give a misleading impression of what they are comparing in the abstract and introduction. These sections do not accurately reflect what was done in the study, as they not only compare Bayesian and Frequentist frameworks, but also different methodologies, including Kriging and Bayesian hierarchical modelling. Kriging is not the same as a Bayesian hierarchical model. While both techniques involve the use of spatial dependence or correlation, they are distinct and serve different purposes. Kriging is a spatial interpolation technique used to estimate values at unobserved locations, while hierarchical modelling is used to model complex data structures with multiple levels of variation. I am therefore not surprised that different methods that make different assumptions yield different results. I would strongly suggest the authors clarify this aspect as this is not a comparison of Bayesian vs. Frequentist.

It appears that in the discussion, the authors keep shifting their focus between comparing different statistical frameworks (Bayesian and Frequentist) and discussing issues related to the use of Remote Sensing (RS) data, specifically the comparison between Landsat 8 and Sentinel 2. While I understand that these topics are interconnected to some extent, I would appreciate more clarity in their presentation.

The authors' discussion could be improved by clearly separating the two topics and providing more explanation on the relationship between them. While statistical frameworks can certainly impact the analysis of RS data, the authors should make sure that their discussion of the two topics is distinct and does not conflate one with the other.

Overall, a clearer and more focused discussion would enhance the discussion and improve the understanding of the audience.

I have also noticed that the authors have not provided the code used for the analyses. In my opinion, it would be beneficial for them to share the code to ensure full reproducibility of the results. Providing the code would allow other researchers to verify and replicate the analyses and results, and potentially build upon the study. Therefore, I would suggest that the authors consider including the code as a supplementary material for the readers.

Author Response

(The authors gave the same response as above.)
